# Yoga programme for type-2 diabetes prevention (YOGA-DP) among high risk people in India: a multicentre feasibility randomised controlled trial protocol

Kaushik Chattopadhyay [ID],[1] Pallavi Mishra,[2] Kavita Singh,[2] Tess Harris [ID],[3] Mark Hamer [ID],[4] Sheila Margaret Greenfield,[5] Sarah Anne Lewis,[1] Nandi Krishnamurthy Manjunath,[6] Rukamani Nair,[7] Somnath Mukherjee,[7] David Ross Harper,[8] Nikhil Tandon,[9] Sanjay Kinra,[10] Dorairaj Prabhakaran,[2] On behalf of YOGA-DP Study Team

For numbered affiliations see end of article.

**Correspondence to**
Dr Kaushik Chattopadhyay;
kaushik.chattopadhyay@
nottingham.ac.uk

## ABSTRACT

**Introduction** A huge population in India is at high risk of type-2 diabetes (T2DM). Physical activity and a healthy diet (healthy lifestyle) improve blood glucose levels in people at high risk of T2DM. However, an unhealthy lifestyle is common among Indians. Yoga covers physical activity and a healthy diet and can help to prevent T2DM. The research question to be addressed by the main randomised controlled trial (RCT) is whether a Yoga programme for T2DM prevention (YOGA-DP) is effective in preventing T2DM among high risk people in India as compared with enhanced standard care. In this current study, we are determining the feasibility of undertaking the main RCT.

**Intervention** YOGA-DP is a structured lifestyle education and exercise programme. The exercise part is based on Yoga and includes Shithilikarana Vyayama (loosening exercises), Surya Namaskar (sun salutation exercises), Asana (Yogic poses), Pranayama (breathing practices) and Dhyana (meditation) and relaxation practices.

**Methods and analysis** This is a multicentre, two-arm, parallel-group, feasibility RCT with blinded outcome assessment and integrated mixed-methods process evaluation. Eligible participants should be aged 18–74 years, at high risk of T2DM (fasting plasma glucose level 5.6–6.9 mmol/L) and safe to participate in physical activities. At least 64 participants will be randomised to intervention or control group with final follow-up at 6 months. Important parameters, needed to design the main RCT, will be estimated, such as SD of the outcome measure (fasting plasma glucose level at 6-month follow-up), recruitment, intervention adherence, follow-up, potential contamination and time needed to conduct the study. Semistructured qualitative interviews will be conducted with up to 20–30 participants, a sample of those declining to participate, four YOGA-DP instructors and around eight study staff to explore their perceptions and experiences of taking part in the study and of the intervention, reasons behind non-participation, experiences of delivering the intervention and running the study, respectively.

### Strengths and limitations of this study

► We are determining the feasibility of undertaking the main randomised controlled trial (RCT), and important parameters, needed to design the main RCT, will be estimated.
► This is a multicentre, two-arm, parallel-group, feasibility RCT with blinded outcome assessment and integrated mixed-methods process evaluation.
► The study is registered with the Clinical Trials Registry-India, a part of the WHO Registry Network.
► Being a feasibility RCT, it is not adequately powered to detect a difference in outcomes between the two study arms.
► However, appropriate regression methods will be used to get initial estimates of effects with CIs to guide the design of the main RCT.

**Ethics and dissemination** Ethics approval has been obtained from the following Research Ethics Committees: Faculty of Medicine and Health Sciences, University of Nottingham (UK); Centre for Chronic Disease Control (CCDC, India); Bapu Nature Cure Hospital and Yogashram (BNCHY, India) and Swami Vivekananda Yoga Anusandhana Samsthana (S-VYASA, India). The results will be widely disseminated among key stakeholders through various avenues.

**Trial registration number** CTRI/2019/05/018893.

## INTRODUCTION

India has the world's second-largest type-2 diabetes (T2DM) epidemic, a disorder with significant health, social and economic consequences.[1] More than 77 million Indians are in the high risk of T2DM category, with higher blood sugar levels than normal, but lower than the established threshold for

T2DM itself.[2] They are more likely to develop T2DM and its complications than people with normal blood glucose levels.[3] Physical inactivity and an unhealthy diet are important risk factors of T2DM.[3] Screening of people at high risk of T2DM, followed by an effective lifestyle intervention (ie, physical activity and a healthy diet) is a cost-effective strategy.[3] It improves blood glucose levels in people at high risk of T2DM and has other health benefits.[4 5] However, physical activity levels are lower among Indians.[6] Similarly, consumption of an unhealthy diet is high among Indians.[7 8]

Yoga, an ancient Indian mind-body discipline, covers not only physical activity, but also a healthy diet.[9] There are many different styles of Yoga, focusing on the same core issue, that is, a healthy lifestyle. No style is necessarily better or more authentic than any other.[10] The acceptability of Yoga is usually high among Indians because it fits their health beliefs and culture.[11 12] Generally, Yoga uses a gentle approach, is easy to learn and safe, requires a low to moderate level of guidance, is inexpensive to maintain and can be practised indoors and outdoors.[11] It can be practised by older people or those with a wide range of comorbidities—it can help with arthritis and can prevent falls.[10 11] Some of the Yogic practices are of low-intensity (<3.5 kcal/min) and some are of moderate-intensity (3.5–7.0 kcal/min).[10 13] For example, the Surya Namaskar component of Yoga (sun salutation exercises) burns about 3.8–6.7 kcal/min.[14 15] Yoga is also considered as a muscle-strengthening activity.[10] Thus, it can contribute to the aim of routine lifestyle advice to prevent T2DM among high risk individuals.

The beneficial effects of Yoga practice on T2DM-related risk profiles appear to occur via two major pathways. First, by reducing the activation and reactivity of the sympathoadrenal system and the hypothalamic-pituitary-adrenal axis, and promoting feelings of well-being, it may alleviate the effects of stress and foster multiple positive downstream effects on neuroendocrine status, metabolic function and related systemic inflammatory responses. Second, by directly stimulating the vagus nerve, it may enhance parasympathetic activity and lead to positive changes in cardiovagal function, mood, energy state and in related neuroendocrine, metabolic and inflammatory responses. Furthermore, Yoga may lead to weight loss, which itself lowers the risk of T2DM.[16]

Systematic reviews of clinical trials suggest beneficial effects of Yoga on T2DM-related outcomes in T2DM (as adjuvant therapy) and in metabolic syndrome.[17–20] One such systematic review of 44 randomised controlled trials (RCTs) analysed data from T2DM, metabolic syndrome and healthy participants (n=3168).[17] Relative to usual care or no intervention, Yoga improved blood glucose levels (mean difference=−0.45%; 95% CI −0.87 to −0.02). No major safety issues were reported. However, most of the included studies were short-term (≤3 months) and were often associated with considerable methodological limitations, such as small sample sizes in treatment groups, resulting in lack of statistical power for outcome assessment, and poor concealment of treatment allocation in outcome assessment, leading to potential analysis bias.

In addition, some of the relevant previous studies have not described the intervention in detail, making it difficult to replicate successful interventions.[17–20] Most studies have not reported the intervention development process. It is hard to know whether these interventions were carefully thought out (eg, their safety and acceptability) and comprehensive in their development. Thus, it is difficult to select (and replicate) one successful intervention over another. A further selection barrier is their heterogeneous contents, which needed to be summarised for utilisation in T2DM prevention. Therefore, we addressed these issues by systematically developing a Yoga programme for T2DM prevention (YOGA-DP) among high risk people in India, which will be published elsewhere. Briefly, this iterative process included five steps: (1) a systematic review of the literature to generate a list of Yogic practices that improves blood glucose levels among adults at high risk of or with T2DM, (2) validation of identified Yogic practices by Yoga experts, (3) development of the intervention, (4) consultation with a range of relevant experts about the intervention and (5) pretest the intervention among Yoga practitioners and lay people in India.

Health interventions should be informed by and compatible with the sociocultural expectations of people and their health beliefs.[21] Yoga is such an intervention in India. The Indian government is committed to and has prioritised the prevention and management of chronic diseases like T2DM through traditional Indian therapies like Yoga. The Ministry of AYUSH is dedicated exclusively towards traditional Indian therapies.[22] There is, therefore, a need for a definitive, robustly designed study to assess the utility of Yoga in T2DM prevention among high risk people in India. The principal research question to be addressed by the main RCT is whether YOGA-DP is effective in preventing T2DM among high risk people in India as compared with enhanced standard care. The primary outcome of the main RCT will be the difference in mean fasting plasma glucose level between the two treatment arms. We intend to do long-term (≥1 year) follow-ups in the main RCT. The chances of successful completion of a costly T2DM prevention RCT will improve if the feasibility of its key elements is checked before it starts.[23 24] Important parameters, needed to design the main RCT, will be estimated.[23] Thus, in this current study, we are determining the feasibility of undertaking the main RCT.

## METHODS AND ANALYSIS
### Study design
This is a multicentre, two-arm, parallel-group, feasibility RCT (see figure 1) with blinded outcome assessment and integrated mixed-methods process evaluation.

### Study setting
The study is conducted at two Yoga centres in India—one each in the northern part of India (Bapu Nature

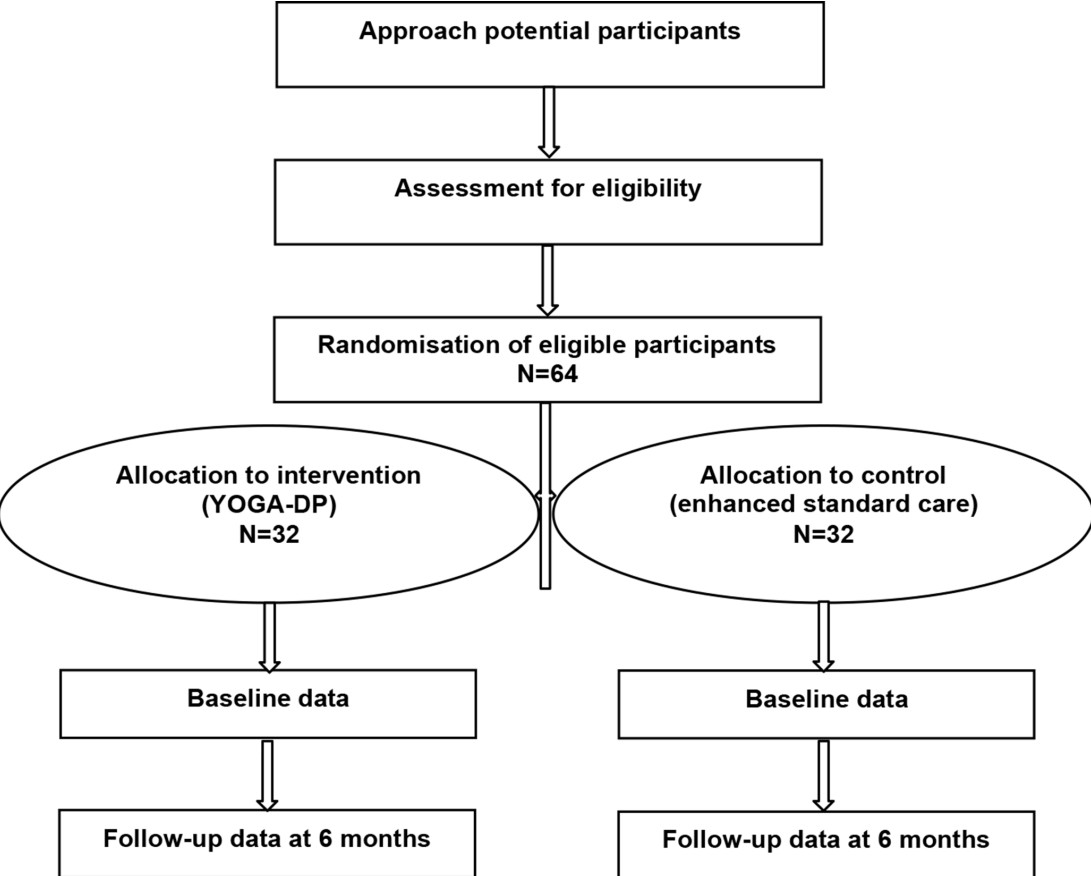

**Figure 1** Randomised controlled trial design. YOGA-DP, Yoga programme for T2DM prevention.

Cure Hospital and Yogashram (BNCHY, New Delhi)) and southern part of India (Swami Vivekananda Yoga Anusandhana Samsthana (S-VYASA, Bengaluru)). People from a range of socioeconomic backgrounds access the services provided by these two research-active Yoga centres. Three languages (English, Hindi and Kannada) are used to conduct the study.

### Sample size estimation
#### Randomised controlled trial
At least 64 participants (32/group) will be adequate to precisely estimate the SD of the outcome measure (fasting plasma glucose level at 6-month follow-up). This is calculated in relation to the desired level of confidence (95%) for the SD, the chosen power (80%) and significance level (5%, two-tailed) of the analysis in the main RCT and the expected loss to follow-up (20% at 6 months) in the current study.[25 26]

#### Qualitative evaluation
► Participants: interviews will be conducted with up to 20–30 participants. Until data saturation is achieved, purposive sampling will be used to ensure the representation of diversity within the RCT population.[27]
► Those who decline to participate in the study: a sample of those who agree to be interviewed about

their reasons for non-participation, around 10–15, but will continue until saturation is reached.[27]
► Four YOGA-DP instructors and around eight study staff (at the two sites).

### Screening and recruitment strategies
A multipronged approach is used to identify potential participants at both sites:
► Advertisement through posters and pamphlets (placed/distributed at various locations including these Yoga centres, communities, religious places, parks and health clinics).
► Screening camps at various locations (including these Yoga centres, communities and religious places) and times.
► Door to door visits in various communities and at various times.

After potential participants have been given the participant information sheet, a description of the study and any questions have been answered, people interested in the study are requested to provide written informed consent. Those providing written informed consent are assessed against the study eligibility criteria. Their fasting blood glucose level is determined by finger-prick using a glucometer. At these two sites, two glucometer brands are used for this purpose: HemoCue Glucose 201⁺ System and Accu-Chek Active. Those potentially at high risk of

**Table 1**  Data collection

|  | Face-to-face assessments* | | | |
|  | Assessment details | Screening and recruitment | Baseline | Final at 6 months |
|---|---|---|---|---|
| Eligibility assessment |  | √ |  |  |
| Socio-demographics |  |  | √ |  |
| Medical and surgical history |  |  | √ |  |
| Family history of diabetes |  |  | √ |  |
| Current medications |  |  | √ | √ |
| Biochemical parameters† |  |  |  |  |
| *Blood glucose* |  |  |  |  |
| Fasting plasma glucose | Glucose oxidase-peroxidase method |  | √ | √ |
| Glycated haemoglobin | High-performance liquid chromatography method |  | √ | √ |
| *Lipid profile* |  |  |  |  |
| Total cholesterol | Cholesterol oxidase method |  | √ | √ |
| High-density lipoprotein | Direct clearance method |  | √ | √ |
| Low-density lipoprotein | Direct clearance method |  | √ | √ |
| Very low-density lipoprotein | Calculated value |  | √ | √ |
| Triglyceride | Lipase/Glycerol-3-phosphate oxidase-phenol+aminophenazone no correction method |  | √ | √ |
| Physiological parameters |  |  |  |  |
| Blood pressure | Omron HEM-7201 |  | √ | √ |
| Heart rate | Omron HEM-7201 |  | √ | √ |
| Anthropometric parameters |  |  |  |  |
| Waist circumference | Seca 201 (measuring tape) |  | √ | √ |
| Weight | Omron HN-286 (weighing scale) |  | √ | √ |
| Height | Seca 206 (stadiometer) |  | √ | √ |
| Body mass index | Calculated value |  | √ | √ |
| Diet | Time-recall: past 1 week |  | √ | √ |
| Physical activity | International Physical Activity Questionnaire—short; time-recall: past 1 week[33] |  | √ | √ |
| Tobacco usage |  |  | √ | √ |
| Alcohol consumption |  |  | √ | √ |
| Health-related quality-of-life | EuroQol-5D-5L; time-recall: at the time of questionnaire completion[34] |  | √ | √ |
| Depression, anxiety and stress | Depression, Anxiety and Stress Scale; time recall: past 1 week[35] |  | √ | √ |
| Yoga practice | Time-recall: past 1 week |  | √ | √ |
| Self-efficacy (to assess confidence in participant's ability to practise Yoga) | 0–100 rating scale; time-recall: at the time of questionnaire completion[36] |  | √ | √ |

*A standard operating procedure has been developed for this purpose.
†Blood samples are analysed at the International Organization for Standardization or Christian Medical College External Quality Assurance Scheme (Vellore, India) accredited laboratories.

T2DM (ie, fasting blood glucose level 5.6–6.9 mmol/L (ie, 100–125 mg/dL))[28] are invited to these Yoga centres for a confirmatory venous blood test, using a standardised method (see table 1) and after taking further written informed consent. They are re-assessed against the eligibility criteria for the study.

## Eligibility criteria
### Inclusion criteria
Participants should be:
► Aged 18–74 years.
► At high risk of T2DM.

► Safe to participate in physical activities (assessed using the Physical Activity Readiness Questionnaire (PAR-Q)/clinician).[29]
► Willing and able to attend the intervention/control sessions on their own.
► Able to provide written informed consent.

### Exclusion criteria
► Pregnant women.
► Those with glycated haemoglobin ≥6.5% (ie, ≥48 mmol/mol; with T2DM).[28]

**Table 2** Structure of YOGA-DP

| Week | Group Yoga sessions delivered by YOGA-DP instructors | Self-practice of Yoga at home using YOGA-DP booklet and a video | Extra features |
|---|---|---|---|
| 1–4 (month 1) | At least two sessions of 45 min per week. An attendance register is kept. | – | At the first session, the instructor is giving participants part one of our programme booklet. This gives them information about being at high risk of T2DM and how to prevent T2DM (ie, by being more physically active, keeping a healthy weight, eating less fat (especially saturated fat) and eating more fibre). |
| 5–12 (month 2–3) | At least two sessions of 75 min per week. An attendance register is kept. | – | At the last session, the instructor is giving participants part two of our programme booklet and a video. These give them information on Yoga practice to prevent T2DM. Also, a Yoga diary and non-slippery Yoga mat are provided for self-practice of Yoga at home. |
| 13–24 (month 4–6) | At least one session of 75 min every 4 weeks. An attendance register is kept. | At least two sessions of 75 min per week. Participants are given the Yoga diary to record their Yoga practice (types and minutes). | The instructor is phoning participants every week to offer support and help and to troubleshoot any problems. |
| 25+ (month 7+) | – | At least two sessions of 75 min per week. Participants are given the Yoga diary to record their Yoga practice (types and minutes). | – |

T2DM, type-2 diabetes; YOGA-DP, Yoga programme for T2DM prevention.

► Those with any serious or uncontrolled medical condition (eg, cancer).
► Those who regularly practice Yoga, that is, ≥150 min/week.
► Those currently receiving (or with plans to receive during the study period) any related non-pharmaceutical/pharmaceutical research intervention.

### Randomisation

Eligible participants are randomised to intervention or control group according to a computer-generated randomisation schedule (1:1, block randomisation, stratified by sex and site), done centrally by an independent statistician at the Centre for Chronic Disease Control (CCDC), New Delhi, India. This is accessed by calling a telephone line. The exception to this rule is individuals recruited from the same household or if they are close relatives or friends, who are randomised to the same group to avoid contamination. After randomisation, key baseline data are collected. Participants and intervention/control providers cannot be 'blinded' to group allocation, but the outcome assessors are 'blind'.

### Interventions

#### Intervention (YOGA-DP)

YOGA-DP is a structured lifestyle education and exercise programme, provided over a period of 24 weeks (see table 2). The exercise part is based on Yoga and includes Shithilikarana Vyayama (loosening exercises), Surya Namaskar (sun salutation exercises), Asana (Yogic poses), Pranayama (breathing practices) and Dhyana (meditation) and relaxation practices. Online supplementary table S1 shows the structure and content of the Yoga sessions. The programme is delivered by YOGA-DP instructors—qualified and experienced Yoga teachers with formal training provided on the intervention. Female instructors are available for female participants. Group Yoga sessions are run locally (such as at these Yoga centres and community centres) at different time points of the day (with evening and weekend sessions), and participants can join as per their convenience. We are reimbursing some of their local travel costs for attending the sessions. A family member or someone close to the participant is invited to join them in the sessions. Once participants complete the programme, they are strongly encouraged to maintain a healthy lifestyle in the long-term, using the intervention booklet and a video.

Intervention fidelity will be ensured through regular training of YOGA-DP instructors, based on the manual developed for them. Also, sessions will be regularly observed and assessed with a checklist to ensure that they are being delivered as per the manual. To improve performance, structured and instructive feedback will be provided.

#### Control (enhanced standard care)

Currently, no standard lifestyle intervention is available in India for people at high risk of T2DM. Control group participants will receive a leaflet on routine lifestyle advice to prevent T2DM among high risk individuals.

This is delivered by a different team member (ie, not by the YOGA-DP instructor) to avoid contamination. This provision would ensure that control group participants feel that there are benefits to participation (hence, lower attrition). Contamination could occur in the control group if they start practising Yoga during follow-up. However, the specific intervention (YOGA-DP) is not available externally, even if Yoga classes are.

### Study parameters and data collection
#### Randomised controlled trial
► SD of the outcome measure (mentioned before), which will be used to calculate the main RCT sample size.
► Recruitment—number of people approached to participate, written informed consent given, screened for eligibility, found eligible and randomised.
► Intervention adherence—number of sessions attended out of the 27 sessions, number who self-practice at home and frequency and duration of self-practice at home.
► Follow-up—number of randomised participants followed-up at 6 months.
► Potential contamination—number of control group participants participating in any Yoga class during 6-month follow-up (self-reported).
► Time needed to conduct the study (eg, to recruit participants).
► See table 1.

#### Qualitative evaluation
► Participants: interviews will be conducted with them to explore their perceptions and experiences of taking part in the study (intervention and control group participants who complete or do not complete the study) and of the intervention (intervention group participants who complete or do not complete the intervention).
► Those who decline to participate in the study: they are requested to complete a questionnaire (including reasons behind non-participation), and a sample of those who agree to be interviewed to further explore these reasons.
► YOGA-DP instructors and study staff (at the two sites): interviews will be conducted with them to explore their experiences of delivering the intervention and running the study, respectively.

Predeveloped interview guides will be used by a qualitative researcher to conduct these semistructured interviews. The interviews will be conducted in interviewees' preferred language and with the help of an interpreter if needed. With consent, these will be noted and digitally recorded.

### Data analyses
#### Randomised controlled trial
Baseline characteristics and important parameters such as follow-up will be summarised and compared between the two study arms using numbers and percentages for categorical data and summary measures of mean or median and spread for continuous data. Being a feasibility RCT, it is not adequately powered to detect a difference in outcomes between the two study arms. However, appropriate regression methods will be used to get initial estimates of effects with CIs to guide the design of the main RCT. All primary analyses will be based on the intention-to-treat principle and will be unadjusted. Subsequently, the adjustment will be done for the baseline data and site. No interim analysis is planned.

#### Qualitative evaluation
All the semistructured interviews will be transcribed (verbatim), translated (if necessary), anonymised and checked for accuracy. An interpretive analysis will be conducted using thematic analysis, using NVivo software. Transcripts will be read and re-read by two qualitative researchers. These researchers will develop the initial codes and will apply initially to a small number of transcripts, enabling further iteration of the thematic index. We will use illustrative non-attributable quotations.[27]

### Patient and public involvement
The research topic was identified and discussed with a Public Engagement Coordinator and among a patient and public involvement group. They acknowledged the importance of this research topic and the issues identified during these discussions were taken into consideration while designing the study. They are involved in the discussions and are giving feedback on different aspects of the study.

## ETHICS AND DISSEMINATION
### Ethics and other related issues
Ethics approval has been obtained from the following Research Ethics Committees: Faculty of Medicine and Health Sciences, University of Nottingham, UK (14-1805); CCDC, India (CCDC_IEC_09_2018); BNCHY, India (BNCHY/IEC/2/2019) and S-VYASA, India (RES/IEC-SVYASA/138/2018). Written informed consent is obtained from all the participants. We have also received approval from the Health Ministry's Screening Committee (HMSC, India). An independent Trial Steering Committee is monitoring and providing overall supervision for the study.

### Serious adverse events
Like other physical activities, Yoga is known to be safe.[10] Information will be collected on serious adverse events (including death) occurring in participants that may be attributed to the interventions. Based on medical and scientific judgement, an independent clinician will determine the relationship of any event to the interventions.

## Participant withdrawal

Participants will be withdrawn from the study either at their request or at the discretion of the site investigator for example, if diagnosed with diabetes (will receive the standard treatment) or if no longer safe to participate in physical activities (determined by PAR-Q/clinician).[29] They will be made aware that this will not affect their future care. Also, they will be made aware (via the participant information sheet and consent form) that should they withdraw, the data collected to date will not be erased and may still be used in the final analyses.

## Dissemination

The results will be reported according to the relevant extension of the Consolidated Standards of Reporting Trials statement, that is, for randomised pilot and feasibility trials.[30] The results will be widely disseminated among key stakeholders through various avenues, such as through dissemination meetings and informal discussions with them, presentations at national and international conferences, publications in peer-reviewed open-access journals and press offices and websites of host institutions.

## DISCUSSION

We are now conducting a multicentre feasibility RCT in India to determine the feasibility of undertaking the main RCT. The study started in May 2019, and we are aiming to finish the study by the end of October 2020. If the feasibility is promising (such as recruitment, randomisation, intervention adherence and follow-up), then the parameters estimated will be used to design the main RCT. Decisions over whether to modify the protocol will be informed by the process evaluation, including the qualitative data.

If the intervention is found to be effective in the main RCT, it will be a low-cost, acceptable and local solution to prevent T2DM among high risk people in India and to become healthier overall. The future clinical, personal and economic burden of T2DM on patients, their families, the health system and the economy will be prevented. The benefits of preventing T2DM may extend to the prevention of its complications. People will be provided with more evidence-based choices for preventing T2DM. The programme will simultaneously empower them to manage their health. Apart from India and neighbouring South Asian countries, Yoga is popular or becoming popular in many other countries.[31 32] Given that T2DM prevention is a global concern and costs are a concern everywhere, a low-cost Yoga-based T2DM prevention option will be of interest in other countries, particularly in other South Asian countries and in countries with South Asian ethnic minorities.

## Author affiliations
[1]Division of Epidemiology and Public Health, University of Nottingham, Nottingham, UK
[2]Centre for Chronic Disease Control, Delhi, India
[3]Population Health Research Institute, St George's University of London, London, UK
[4]Institute Sport Exercise and Health, Division of Surgery and Interventional Science, University College London, London, UK
[5]Institute of Applied Health Research, University of Birmingham, Birmingham, UK
[6]Swami Vivekananda Yoga Anusandhana Samsthana, Bengaluru, India
[7]Bapu Nature Cure Hospital and Yogashram, Delhi, India
[8]Harper Public Health Consulting Limited, London, UK
[9]All India Institute of Medical Sciences, Delhi, India
[10]London School of Hygiene And Tropical Medicine, London, UK

**Acknowledgements** The authors would like to extend thanks to all who have participated in this study and the TSC members.

**Contributors** KC conceptualised and designed the study with the help of TH, MH, SMG, SAL, NKM, DRH, NT, SK and DP. KC wrote the first draft of the manuscript. PM, KS, TH, MH, SMG, SAL, NKM, RN, SM, DRH, NT, SK and DP contributed significantly to the revision of the manuscript. All authors read and approved the final manuscript.

**Funding** The study is funded by the UK's DFID/MRC/NIHR/Wellcome Trust Joint Global Health Trials (MR/R018278/1). The funding agencies have no role in designing the study or in writing the manuscript.

**Competing interests** None declared.

**Patient and public involvement** Patients and/or the public were involved in the design, or conduct, or reporting, or dissemination plans of this research. Refer to the Methods section for further details.

**Patient consent for publication** Not required.

**Provenance and peer review** Not commissioned; externally peer reviewed.

**ORCID iDs**
Kaushik Chattopadhyay http://orcid.org/0000-0002-3235-8168
Tess Harris http://orcid.org/0000-0002-8671-1553
Mark Hamer http://orcid.org/0000-0002-8726-7992

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
