## [Reviewer comments · BMJ Open]

ARTICLE DETAILS

TITLE (PROVISIONAL)	Yoga programme for type-2 diabetes prevention (YOGA-DP) among high risk people in India: a multi-centre feasibility randomised controlled trial protocol
AUTHORS	Chattopadhyay, Kaushik; Mishra, Pallavi; Singh, Kavita; Harris, Tess; Hamer, Mark; Greenfield, Sheila; Lewis, Sarah; Manjunath, NK; Nair, Rukamani; Mukherjee, Somnath; Harper, David; Tandon, Nikhil; Kinra, Sanjay; Prabhakaran, Dorairaj

VERSION 1 – REVIEW

REVIEWER	SATISH PATIL Shri B M Patil Medical College Hospital & Research Centre, BLDE (Deemed to be University), India
REVIEW RETURNED	06-Mar-2020

GENERAL COMMENTS	Comments: The objective of the proposed protocol is to determine the feasibility of undertaking the randomized controlled trial of yoga programme on type-2-diabetes prevention among high risk people in India. This is an ongoing multi-centre, two-arm, parallel-group, feasibility RCT. The outcome assessment will be blinded. The study has been registered in the clinical trial registry-India. It is a good practice to conduct the feasibility trial before conducting the main RCT. The outcomes of this study will guide in designing the RCT appropriately. The study design is good and well-planned. The protocol has been prepared systematically. Minor issues: Few minor corrections could improve the quality of protocol. Abstract & Method: I suggest giving fixed number of participants from whom semi-structured qualitative data will be collected (instead of 20-30 participants).As the subjects will be screened from two large regions of the India (Northern and southern part of India), mention the sampling method.Intervention (Yoga-DP) programme needs to be elaborated. Give details of yoga sessions (45 minute and 75 minute).Ethics approval and consent has been mentioned twice: mention it in methodology or declaration section.Sample size estimation has been given in between the data collection. Replace sample size estimation from the data collection to after study setting paragraph.
--

REVIEWER	Akshay Anand Post Graduate Institute of Medical Education and Research
REVIEW RETURNED	26-Mar-2020

GENERAL COMMENTS	The proposed study is about determining the feasibility of yoga programme for T2DM prevention (YOGA-DP) and its effectiveness in prevention of T2DM among high risk people in India as compared to enhanced standard care by addressing main randomised controlled trial. The plan is quite wonderful as researchers are trying to utilize yogic techniques in order to alleviate the T2DM in effective and more scientific manner. I have several question regarding this study, which are as follows:  1. Mention about yoga protocol, which is to be used in the study in abstract section. 2. Please mention consort statement 3. The main purpose of the study is know whether a yoga programme for T2DM prevention (YOGA-DP) is effective in preventing T2DM among high risk people in India as compared to enhanced standard care. What is the significance of using YOGA-DP for high risk individual in comparison to conventional yoga techniques? 4. What is the basis of selecting two centres for Yoga and why? 5. As you mentioned that 'The intervention has been systematically developed by our study team through reviewing the scientific literature and in consultation with a range of stakeholders (including healthcare, medical and yoga experts and practitioners and the public), which will be published elsewhere'. Please define how have you developed this yoga protocol for high risk of T2DM individuals 6. What would be the measures for the compliance of participants in yoga protocol? 7. What are the measures you will be taking to control their diet, as this is the one important factor, which can give false positive outcomes? 8. Schematic of study design/strategy is required 9. Please provide ethical reference number. 10. The age included for the study is very wide i.e. 18-74 years. How will you going to address age related outcomes as well as variation?^[SEP] 11. You have calculated 64 participants for the study. Have you also consider dropouts in this?
--

VERSION 1 – AUTHOR RESPONSE

Reviewer: 1

The objective of the proposed protocol is to determine the feasibility of undertaking the randomized controlled trial of yoga programme on type-2-diabetes prevention among high risk people in India. This is an ongoing multi-centre, two-arm, parallel-group, feasibility RCT. The outcome assessment will be blinded. The study has been registered in the clinical trial registry-India. It is a good practice to conduct the feasibility trial before conducting the main RCT. The outcomes of this study will guide in designing the RCT appropriately. The study design is good and well-planned. The protocol has been prepared systematically. Minor issues: Few minor corrections could improve the quality of protocol.

Response: Thank you very much for reviewing our manuscript and appreciating the work.

Abstract & Method:

a. I suggest giving fixed number of participants from whom semi-structured qualitative data will be collected (instead of 20-30 participants).

Response: Thank you for your suggestion. As this part is a qualitative study, we need to continuously conduct and analyse interviews until data saturation is achieved. Thus, we cannot say the exact number at this point in time. We are hoping to achieve data saturation with up to 20-30 interviews. We have given a potential maximum sample size based on existing qualitative methods literature and it also reflects our experience in previous studies. It is in accordance with recommended sample sizes which typically allow for data saturation to be reached within this type of study – it allows leeway for further participants to be interviewed if it appears data saturation may have been reached but it would be helpful to confirm by checking with a few more participants. Just to reassure, we have two trained and experienced qualitative researchers in our study team, including Prof. Sheila Margaret Greenfield at the University of Birmingham, UK. Prof. Greenfield, a qualitative methodologist, has research experience in complementary therapies used for managing chronic conditions and planning and undertaking qualitative evaluations.

b. As the subjects will be screened from two large regions of the India (Northern and southern part of India), mention the sampling method.

Response: The same sampling method is used at both sites. We have mentioned the following in the manuscript and have added a comment to the manuscript that the same sampling method is used at each site, for clarity-

“Screening and recruitment strategies

A multipronged approach is used to identify potential participants at both sites:

- Advertisement through posters and pamphlets (placed/distributed at various locations including these Yoga centres, communities, religious places, parks and health clinics).
- Screening camps at various locations (including these Yoga centres, communities and religious places) and times.
- Door to door visits in various communities and at various times.

After potential participants have been given the participant information sheet, a description of the study and any questions have been answered, people interested in the study are requested to provide written informed consent. Those providing written informed consent are assessed against the study eligibility criteria. Their fasting blood glucose level is determined by finger-prick using a glucometer. At these two sites, two glucometer brands are used for this purpose: HemoCue Glucose 201⁺ System and Accu-Chek Active. Those potentially at high risk of T2DM (i.e., fasting blood glucose level 5.6 to 6.9 mmol/L (i.e., 100 to 125 mg/dL)) [26] are invited to these Yoga centres for a confirmatory venous blood test, using a standardised method (see Table 1) and after taking further written informed consent. They are re-assessed against the eligibility criteria for the study.

Randomisation

Eligible participants are randomised to intervention or control group according to a computer-generated randomisation schedule (1:1, block randomisation, stratified by sex and site), done centrally by an independent statistician at the Centre for Chronic Disease Control (CCDC), New Delhi, India. This is accessed by calling a telephone line. The exception to this rule is individuals recruited from the same household or if they are close relatives or friends, who are randomised to the same group to avoid contamination. After randomisation, key baseline data are collected. Participants and intervention/control providers cannot be 'blinded' to group allocation but the outcome assessors are 'blind'."

Please let us know if you want us to provide any additional information.

c. Intervention (Yoga-DP) programme needs to be elaborated. Give details of yoga sessions (45 minute and 75 minute).

Response: We have described this in the intervention development manuscript. Having said that and as suggested, we have added the following brief (due to the word limit) information in the abstract-

Intervention YOGA-DP is a structured lifestyle education and exercise programme. The exercise part is based on Yoga and includes Shithilikarana Vyayama (*loosening exercises*), Surya Namaskar (sun salutation exercises), Asana (Yogic poses), Pranayama (breathing practices) and Dhyana (meditation) and relaxation practices.

In the methods section, we have added the following information-

Table S1 shows the structure and content of the Yoga sessions.

Table S1: Structure and content of the Yoga sessions

Yogic practices	Week 1-4	Week 5+	Details
	Each session will last for 45 minutes with the time split as follows:	Each session should last for 75 minutes with the time split as follows:	
Shithilikarana Vyayama	Around 5 minutes	Around 5 minutes	(1) Neck rotation 30 seconds (2) Shoulder rotation 30 seconds (3) Elbow flexion and extension 30 seconds

			(4) Wrist rotation 30 seconds (5) Finger movement 30 seconds (6) Waist rotation 30 seconds (7) Knee flexion and extension 1 minute (8) Ankle rotation 1 minute (9) Toe movement 30 seconds
Surya Namaskar	--	Around 15 minutes	The below mentioned 12 steps constitute one set of Surya Namaskar. To complete one round of Surya Namaskar, participants need to repeat these 12 steps on the other side of their body (i.e., by extending their left leg behind in step number 4 and bringing their left leg forward in step number 9). Initially, they should practise Surya Namaskar at a slower pace. Only with practice over some time, they may try to do 12 rounds of it at a faster pace for around 15 minutes (i.e., a couple of seconds per step). (1) Pranamasana (prayer pose) (2) Hastauttanasana (raised arms pose) (3) Padahastanasana (hands to feet pose) (4) Ashwa Sanchalanasana (equestrian pose) (5) Dandasana (stick pose) (6) Ashtanga Namaskara Asana (salute with eight parts) (7) Bhujangasana (cobra pose) (8) Parvatasana (mountain pose) (9) Ashwa Sanchalanasana (equestrian pose) (10) Padahastanasana (hands to feet pose) (11) Hastauttanasana (raised arms pose) (12) Pranamasana (prayer pose)
Asana	Around 15 minutes	Around 30 minutes	Two-sided poses (right and left) are to be practised for about 3 minutes (1.5 minutes on each side) and central-positioned poses are to be practised for about 1.5 minutes. In each session, the Yogic poses are selected from the list below to prevent boredom from the similarity of routine. Advanced Yogic poses are introduced from week 5 onwards, for example, Konasana (angle pose), Trikonasana (triangle pose), Paravakonasana (lateral angle pose), Ardhastrasana (half camel pose), Ustrasana (camel pose), Dhanurasana (bow pose) and Naukasana (boat pose). (A) Standing poses

			(1) Tadasana (palm tree pose) 1.5 minutes (2) Ardhashakrasana (half wheel pose) 1.5 minutes (3) Katichakrasana (waist wheel pose) 3 minutes (4) Konasana (angle pose) or Trikonasana (triangle pose) or Paravakonasana (lateral angle pose): alternatively 3 minutes (B) Sitting poses (1) Vajrasana (adamant pose) 1.5 minutes (2) Mandukasana (frog pose) 1.5 minutes (3) Ardhastrasana (half camel pose) or Ustrasana (camel pose): alternatively 1.5 minutes (4) Vakrasana (twisted pose) or Ardhamatsyendrasana (half spinal twist pose): alternatively 3 minutes (5) Paschimottanasana (seated forward bend pose) or Janusirsasana (head to knee pose): alternatively 1.5 minutes or 3 minutes, respectively (C) Lying poses- front/prone (1) Ardhashalabhasana (half locust pose) or Poornashalabhasana (full locust pose): alternatively 3 minutes or 1.5 minutes, respectively (2) Dhanurasana (bow pose) 1.5 minutes (3) Makarasana (crocodile pose) 1.5 minutes (D) Lying poses- back/supine (1) Uttanapadasana (raised legs pose) or Ardhalasana (half plough pose): alternatively 1.5 minutes (2) Pavanamuktasana (wind relieving pose) 1.5 minutes (3) Naukasana (boat pose) 1.5 minutes (4) Saralmatsyasana (easy fish pose) 1.5 minutes
Pranayama	Around 13 minutes	Around 13 minutes	(1) Vibhagiya Pranayama (sectional breathing) 4 minutes (2) Nadishodhana Pranayama (alternate nostril breathing) 3 minutes (3) Kapalbhata Pranayama (skull shining breathing) or Bhastrika Pranayama (bellow breathing): alternately 3 minutes

			(4) Bhramari Pranayama (bee breathing) 3 minutes
Dhyana and relaxation practices	Around 12 minutes	Around 12 minutes	In each session, the following Dhyana and relaxation practices are to be done in a darkened room. (1) A Kara chanting, U Kara chanting and M Kara chanting 3 minutes (2) Yoga Nidra (Yogic sleep) 9 minutes

Additionally, we have now added the following paragraph in the introduction section-

In addition, some of the relevant previous studies have not described the intervention in detail, making it difficult to replicate successful interventions. Most studies have not reported the intervention development process. It is hard to know whether these interventions were carefully thought out (e.g., their safety and acceptability) and comprehensive in their development. Thus, it is difficult to select (and replicate) one successful intervention over another. A further selection barrier is their heterogeneous contents, which needed to be summarised for utilisation in T2DM prevention. Therefore, we addressed these issues by systematically developing a Yoga programme for T2DM prevention (YOGA-DP) among high risk people in India, which will be published elsewhere. Briefly, this iterative process included five steps: (i) a systematic review of the literature to generate a list of Yogic practices that improves blood glucose levels among adults at high risk of or with T2DM, (ii) validation of identified Yogic practices by Yoga experts, (iii) development of the intervention, (iv) consultation with a range of relevant experts about the intervention and (v) pretest the intervention among Yoga practitioners and lay people in India.

d. Ethics approval and consent has been mentioned twice: mention it in methodology or declaration section.

Response: As suggested, we have removed it from the declaration.

e. Sample size estimation has been given in between the data collection. Replace sample size estimation from the data collection to after study setting paragraph.

Response: As suggested, we have amended the manuscript.

Reviewer: 2

The proposed study is about determining the feasibility of yoga programme for T2DM prevention (YOGA-DP) and its effectiveness in prevention of T2DM among high risk people in India as compared to enhanced standard care by addressing main randomised controlled trial. The plan is quite wonderful

as researchers are trying to utilize yogic techniques in order to alleviate the T2DM in effective and more scientific manner. I have several question regarding this study, which are as follows:

Response: Thank you very much for reviewing our manuscript and appreciating the work.

1. Mention about yoga protocol, which is to be used in the study in abstract section.

Response: As suggested, we have added the following brief (due to the word limit) information in the abstract-

Intervention YOGA-DP is a structured lifestyle education and exercise programme. The exercise part is based on Yoga and includes Shithilikarana Vyayama (*loosening exercises*), Surya Namaskar (sun salutation exercises), Asana (Yogic poses), Pranayama (breathing practices) and Dhyana (meditation) and relaxation practices.

2. Please mention consort statement.

Response: As suggested, we have added the following sentence in the manuscript-

The results will be reported according to the relevant extension of the Consolidated Standards of Reporting Trials (CONSORT) statement i.e., for randomised pilot and feasibility trials.

Eldridge SM, Chan CL, Campbell MJ, et al. CONSORT 2010 statement: extension to randomised pilot and feasibility trials. *BMJ*. 2016;355:i5239.

3. The main purpose of the study is to know whether a yoga programme for T2DM prevention (YOGA-DP) is effective in preventing T2DM among high risk people in India as compared to enhanced standard care. What is the significance of using YOGA-DP for high risk individual in comparison to conventional yoga techniques?

Response: We have now added the following paragraph in the introduction section-

In addition, some of the relevant previous studies have not described the intervention in detail, making it difficult to replicate successful interventions. Most studies have not reported the intervention development process. It is hard to know whether these interventions were carefully thought out (e.g., their safety and acceptability) and comprehensive in their development. Thus, it is difficult to select (and replicate) one successful intervention over another. A further selection barrier is their heterogeneous contents, which needed to be summarised for utilisation in T2DM prevention. Therefore, we addressed these issues by systematically developing a Yoga programme for T2DM prevention (YOGA-DP) among high risk people in India, which will be published elsewhere. Briefly, this iterative process included five steps: (i) a systematic review of the literature to generate a list of Yogic practices that improves blood glucose levels among adults at high risk of or with T2DM, (ii) validation of identified Yogic practices by Yoga experts, (iii) development of the intervention, (iv) consultation with a range of relevant experts about the intervention and (v) pretest the intervention among Yoga practitioners and lay people in India.

4. What is the basis of selecting two centres for Yoga and why?

Response: We have mentioned the following in the manuscript-

“The chances of successful completion of a costly T2DM prevention RCT will improve if the feasibility of its key elements is checked before it starts”.

Therefore, if the feasibility is promising, the plan is to conduct the future main RCT at these two sites.

We have now added the following information in the manuscript-

“People from a range of socio-economic backgrounds access the services provided by these two research-active Yoga centres”.

5. As you mentioned that ‘The intervention has been systematically developed by our study team through reviewing the scientific literature and in consultation with a range of stakeholders (including healthcare, medical and yoga experts and practitioners and the public), which will be published elsewhere’. Please define how have you developed this yoga protocol for high risk of T2DM individuals.

Response: We have now amended the above sentence and moved it to the introduction section (please see point number 3 above).

6. What would be the measures for the compliance of participants in yoga protocol?

Response: As mentioned in the manuscript (Table 2), we are using the following-

1. For group Yoga sessions (delivered by the instructor), an attendance register is kept.
2. From self-practice of Yoga at home (using a booklet and video), participants are given a Yoga diary to record their Yoga practice (types and minutes).

We have also mentioned the following under the study parameters and data collection-

“Intervention adherence - number of sessions attended out of the 27 sessions, number who self-practice at home, and frequency and duration of self-practice at home”.

7. What are the measures you will be taking to control their diet, as this is the one important factor, which can give false positive outcomes?

Response: As mentioned in the manuscript, we have used a pragmatic approach and mimicked the real-life situation - in India, the concept of Yoga covers not only physical activity but also a healthy diet. Thus, YOGA-DP is a structured lifestyle education and exercise programme. The exercise part is based on Yoga. In the lifestyle education part, we are covering the following (a booklet is provided to the participants): being at high risk of T2DM and how to prevent T2DM (i.e., by being more physically active, keeping a healthy weight, eating less fat (especially saturated fat) and eating more fibre).

As mentioned in the manuscript, we are also collecting data on their physical activity and diet at baseline and at 6 months.

8. Schematic of study design/strategy is required.

Response: As suggested, we have added this information in the manuscript-

Figure 1: RCT design

9. Please provide ethical reference number.

Response: As suggested, we have now added all the reference numbers-

Ethics approval has been obtained from the following Research Ethics Committees: Faculty of Medicine and Health Sciences, University of Nottingham, UK (14-1805); CCDC, India (CCDC_IEC_09_2018); BNCHY, India (BNCHY/IEC/2/2019) and S-VYASA, India (RES/IEC-SVYASA/138/2018).

10. The age included for the study is very wide i.e. 18-74 years. How will you going to address age related outcomes as well as variation?

Response: As this is a feasibility study, we have made it more inclusive and pragmatic and mimicked the real-life situation by including adults aged 18-74 years. The UK's Joint Global Health Trials (DFID/MRC/NIHR/Wellcome Trust) have given us funds based on this understanding. The aim is to determine the feasibility of undertaking the future main RCT. The main RCT will be adequately powered to detect a difference in outcomes between the two study arms. In this feasibility study, we will assess the feasibility of including this age range in the main RCT. If we include a wide age range in the main RCT, we will consider the potential for any effect to be modified by age, which will be tested by a test for interaction. Prof. Sarah Lewis at the University of Nottingham, UK, is a highly experienced medical statistician. She is part of our study team and is providing statistical input.

There is a potential for confounding if there are differences in age distribution between the two study arms. Randomisation is intended to balance the two groups with respect to age – but we will examine whether there is a difference in age between the two study arms and adjust for this if appropriate.

We have mentioned the following in the manuscript- "Being a feasibility study, it is not adequately powered to detect a difference in outcomes between the two study arms. However, appropriate regression methods will be used to get initial estimates of effects with confidence intervals to guide the design of the main RCT. All primary analyses will be based on the intention-to-treat principle and will be unadjusted. Subsequently, the adjustment will be done for the baseline data and site."

11. You have calculated 64 participants for the study. Have you also considered dropouts in this?

Response: Yes, as mentioned in the manuscript (under sample size estimation), we have taken dropouts (20% at 6-month) into consideration while calculating the sample size.

VERSION 2 – REVIEW

REVIEWER	Satish G Patil Shri B M Patil Medical College Hospital and Research Centre, BLDE Deemed to be University India
-----------------	--

REVIEW RETURNED	21-May-2020
-------------

GENERAL COMMENTS	All the comments has been revised.
------------------------------------